# Cinnamon Oil Alleviates Acetaminophen-Induced Uterine Toxicity in Rats by Abrogation of Oxidative Stress, Apoptosis, and Inflammation

**DOI:** 10.3390/plants12122290

**Published:** 2023-06-12

**Authors:** Sohail Hussain, Saeed Alshahrani, Rahimullah Siddiqui, Andleeb Khan, Manal Mohammed Elhassan Taha, Rayan A. Ahmed, Abdulmajeed M. Jali, Marwa Qadri, Khairat H. M. Khairat, Mohammad Ashafaq

**Affiliations:** 1Department of Pharmacology and Toxicology, College of Pharmacy, Jazan University, Jazan 45142, Saudi Arabiakhk1413@hotmail.com (K.H.M.K.);; 2Substance Abuse Research Center (SARC), College of Pharmacy, Jazan University, Jazan 45142, Saudi Arabia

**Keywords:** acetaminophen, cinnamon oil, uterine toxicity, oxidative stress, inflammation and apoptosis

## Abstract

Paracetamol, or acetaminophen (APAP), is one of the first-line medications that is used for fever and pain. However, APAP can induce uterine toxicity when overused. The mode of action of APAP toxicity is due to the production of free radicals. The main goal of our study is to determine uterine toxicity from APAP overdose and the antioxidative activity of cinnamon oil (CO) in female rats. The effect of different doses of CO (50–200 mg/kg b.w.) was assessed in the uterus toxicity induced by APAP. Additionally, the imbalance in oxidative parameters, interleukins, and caspases was evaluated for the protective effects of CO. A single dose of APAP (2 g/kg b.w.) resulted in uterus toxicity, indicated by a significant increase in the level of lipid peroxidation (LPO), inflammatory interleukins cytokines (IL-1 and 6), expression of caspases 3 and 9, and a marked change in uterus tissue architecture evaluated by histopathology. Co-treatment of CO resulted in a significant amelioration of all the parameters such as LPO, interleukins IL-1β, IL-6, caspases 3 and 9 expression, and distortion of tissue architecture in a dose-dependent manner. Therefore, we can conclude that APAP-induced uterine injury due to oxidative stress can be restored by co-treatment with cinnamon oil (CO).

## 1. Introduction

Paracetamol, also known as acetaminophen and having the IUPAC name N-acetyl- p-aminophenol (APAP), is a widely used drug for pain and fever [1]. It is an organic acid slightly dissolved in water and a weak acid, having a pKa value around 9.5, and therefore, at the physiological range of pH, it remains largely unionized [2]. When APAP is used according to prescribed doses, it is more effective and nontoxic. However, long-term use or excessive doses cause toxic effects in the liver [3], kidney [4], and brain [5]. This toxic effect can be caused by different routes of administration; however, oral administration is the main and most common route of administration among individuals. Passive transport is used for absorption from the gastrointestinal tract [6], and the gastrointestinal mucosa of the rat barely metabolizes the APAP [7]. The rate of absorption depends on how quickly the stomach empties in men since paracetamol is only minimally absorbed from the stomach but rapidly absorbed from the small intestine [8]. Peak plasma concentrations of APAP frequently occur 15 to 30 min after administration of APAP in solution during fasting. Although tablet absorption is typically slower, plasma concentrations are achieved 1 h after administration [9].

Although the pathogenesis of APAP-induced toxicity is still poorly understood, toxic metabolite formation in phase I of drug metabolism is a major contributor because it frequently inhibits mitochondrial function and causes an accumulation of reactive oxygen species (ROS), which causes oxidative stress. The inability of a biological system to quickly detoxify the reactive intermediates or repair the damage that results from them is what is meant by oxidative stress. Several pathological alterations that have been identified, including a wide range of liver and kidney illnesses, depend on this process. The metabolism of APAP changes to toxic N-acetyl parabenzoquinoneimine (NAPQI) in the liver [10], kidney [11], and brain [12] enzymes (Cyp-P450), which produce protein adducts that cause the damage. Therefore, a non-toxic radical formed by glutathione conjugating with NAPQI is excreted from the body [10]. Moreover, an overdose of APAP leads to depletion of GSH, which results in excess production of NAPQI, causing cellular damage. The proper use of antioxidants as a therapeutic strategy for APAP toxicity can result from a deeper understanding of the function of oxidative stress in diseases.

The severity of these organ injuries is associated with the binding of proteins to free radicals [13]. Along with oxidative stress, APAP also activates inflammatory cells, which leads to an increase in the expression of cytokines such as IL-1β and IL-6. Previous studies also proved that an increase in oxidative stress and inflammatory cytokines enhanced APAP-induced toxicity in organs such as the liver [14], kidney [15], and brain [16]. Researchers also showed that APAP causes damage to mitochondrial membrane potential Δψm which activates cytochrome c and caspases 3 and 9, ending in apoptotic cell death [14,16].

Even after modern and advanced medicine were discovered, there is still not a single medicine that is 100% safe, has no side effects, is not toxic to vital organs, or helps in the regeneration of damaged tissues. Furthermore, some drugs may induce toxicity in several organs, causing undesirable side effects. Therefore, to cure diseases more effectively and with less toxicity, it is vital to find alternative medications. Natural antioxidant products, particularly phytochemicals, are extensively utilized due to the involvement of oxidative stress in the etiology of a wide range of disorders. Natural compounds or their derivatives make up close to half of the medications utilized in toxicity therapy today.

Cinnamon is mainly used in flavoring and additives for the food industry. It contains a volatile oil of about 1–2% called cassia oil. Cinnamaldehyde is the main component, with approximately 65–80% of the essential oil and eugenol in lesser amounts. Other components are also found in trace amounts, including starch, mucilage, and tannins [17]. Its bark possesses antibacterial properties and is used as a stomachic, carminative, and antidiarrheal [18]. It has antioxidant properties by scavenging free radicals, antimicrobial properties by toxic action against bacterial cell walls, anti-inflammatory properties through the potent inhibition of nitric oxide (NO), and antimutagenic properties by its modulator effect on the bioactivation and detoxification processes of xenobiotics. Moreover, the central nervous system (CNS) activity of cinnamon was reported to have an anxiolytic effect by regulating the serotonergic and GABAergic systems without any effect on locomotor activity. Additionally, it prevents neuronal cell death through the inhibition of Ca^2+^ influx and prevents diabetes, as was shown by evaluating the effects of its bark extracts on blood glucose and plasma insulin levels in rats under various conditions [19,20]. Its antibacterial activity is likely due to its bacteriostatic ability and toxicity to bacterial cytoplasmic membranes. Moreover, there are nematicidal, immunomodulatory, and allergenic actions, and skin-whitening activities have been studied. It has also been reported to be effective for the treatment of oral cavity infections. Murcia et al. (2004) have shown the antioxidative potency of cinnamon oil (CO) as compared with synthesized food antioxidants, butylated hydroxyl anisole (BHA) (E-320) [21]. CO also showed inhibition of oxidation as determined by the lipid peroxidation assay [22,23]. 

Up until now, few studies have been conducted to show the toxic effect of APAP on pregnant rats [24] and its effect on offspring. Mostly, the studies of APAP were conducted on a human sample during pregnancy for clinical research [25,26]. Our previous reports also showed the protective effect of CO in many organs, like the brain, kidney, and liver. Hence, the current study is intended to investigate the possible protective effects of CO against uterine toxicity by APAP. The toxicity was determined by the evaluation of antioxidants (LPO and GSH), antioxidative enzyme activity, inflammatory cytokines, and caspases by Western blot and histopathological studies.

## 2. Results

### 2.1. Effect of CO on LPO 

A significant rise in LPO in terms of MDA (*p* < 0.01) was obtained in the APAP-treated group III as compared with groups 1 and 2, which correspond to control and CO, respectively. APAP + CO (200 mg) group marked amelioration of the elevated content of MDA (*p* < 0.01) as compared to APAP alone (Figure 1). However, groups APAP + CO (50 mg) and APAP + CO (100 mg) showed insignificant effects.

### 2.2. Effect of CO on GSH Level 

GSH has a major role in the metabolism of APAP and neutralizing the toxic metabolite NAPQI into non-toxic compounds. The level of GSH is lower in the uterine tissue in the APAP group as compared with the control (vehicle only) (*p* < 0.001). Nevertheless, the depleted level of GSH was significantly reversed in animals treated with CO at a high dose group of APAP + CO (200 mg) (*p* < 0.01) as compared with the APAP group (Figure 2). While groups APAP + CO (50 mg) and APAP + CO (100 mg) effects were insignificant.

### 2.3. CO Treatment Protected the Activities of Superoxide Dismutase (SOD) and Catalase Enzymes in the Uterus

After GSH, glutathione-dependent enzymes and antioxidant enzymes have a key role in protecting cells and tissues from APAP toxicity. Due to APAP treatment, a significant decrease in superoxide dismutase (SOD) and catalase (*p* < 0.001 and *p* < 0.01) was observed in only the APAP-treated group (an estimated decline of 56.59% and 39.55%, respectively, was calculated). Nonetheless, co-treatment with CO (200 mg) showed marked improvement in the values of SOD and catalase (*p* < 0.01) (a rise of 84.73% and 57.62% were observed, respectively) as compared to the APAP-treated group (Table 1). Other groups treated with CO (50 and 100 mg) did not reach a significant rise.

### 2.4. CO Treatment Protected the Activities of Glutathione Peroxidase (GPx) and Glutathione Reductase (GR) Enzymes in the Uterus

APAP treatment resulted in downregulation of GPx and GR in the APAP-treated group by 57.37% and 45.52%, respectively, as compared with the control group (*p* < 0.001) (Table 1). Co-treatment of CO (200 mg) showed significant protection of GPx and GR (80.57% (*p* < 0.01) and 57.04% (*p* < 0.001), respectively) as compared with the APAP group. However, co-treatment with lower doses of CO did not significantly enhance GPx and GR. 

### 2.5. Role of CO on APAP-Induced Toxicity in Proinflammatory Cytokines

The treatment of APAP in rats activated the interleukin cytokines IL-1β and IL-6 compared to the control (*p* < 0.001). Furthermore, co-treatment with CO (200 mg) significantly (*p* < 0.01) decreased interleukins levels when compared to the APAP (Figure 3 and Figure 4). However, co-treatment with lower doses of CO did not significantly decrease interleukin levels. 

### 2.6. CO Treatment Suppresses Expression of Activated Caspases 3 and 9

Caspases (3 and 9) expression was significantly increased (*p* < 0.001) in the APAP-treated group when compared with the control. Expression of caspases 3 and 9 was suppressed significantly when treated with CO in the APAP + CO (100 and 200) group as compared with APAP administered (*p* < 0.05 and *p* < 0.01) (Figure 5 and Figure 6). Furthermore, this result was confirmed by conducting a Western blot to determine the expression levels of these two proteins. It was found that the expression of activated caspase 3 was not found in the control group, whereas in the APAP group, the expression was significantly higher as compared with the control group. CO treatment at both 100 and 200 mg doses significantly reduced the expression of caspase 3 band strength as compared to the APAP group. Similar findings were found in caspase 9 expression, where the APAP group showed a broad band as compared with the control, which was significantly and dose-dependently restored by CO. β-actin was used as the standard reference in this experiment (Figure 7).

### 2.7. Effect of CO Treatment on Histopathology of the Uterus

The control group showed normal uterine histology (Figure 8A–D). The regular columnar epithelial lining of the lumen and glands was found to have endometrial columnar cells with an oval nucleus. No cellular changes like fibrosis, apoptosis, or necrosis were visible (A). Treatment with APAP caused histopathological alterations where apoptosis was found in luminal epithelial cells. The glandular epithelial cells underwent degeneration subject to the treatment of APAP (B). There was significant restoration with the treatment of CO in the APAP + CO (100 and 200 mg/kg) group as compared with APAP (C,D). 

## 3. Materials and Methods

### 3.1. Chemicals 

Commercially available acetaminophen (paracetamol) and cinnamon oil were sourced from Sigma Chemicals, Balcatta, WA, USA. All other chemicals and reagents like oxidized glutathione (GSSG), reduced glutathione (GSH), glutathione reductase (GR), nicotinamide adenine dinucleotide phosphate (NADPH), 1-chloro-2,4-dinitrobenzene (CDNB), 5–50 -dithio-bis-2-nitrobenzoic acid (DTNB), thiobarbituric acid (TBA), trichloroacetic acid (TCA), 2,3,5-triphenyltetrazolium chloride (TTC), ethylene diamine tetra-acetic acid (EDTA), (-)epinephrine, paraformaldehyde, glycine, poly-L-lysine, cresyl violet, and catechin hydrate (CH) of the highest analytical grade were purchased from Sigma Chemicals, Balcatta, WA, USA. Cytokines Interleukins Assay Kits of IL-6, IL-β (ab100772, ab100768), caspases 3 and 9 (ab39401 and ab65608), and all antibodies for Western blotting were purchased from Abcam (Cambridge, UK) and Sigma (Sigma chemicals, Balcatta, WA, USA). 

### 3.2. Experimental Animals

Female Wistar rats weighing 200–240 g were acquired from the animal house under the Medical Research Center (MRC) at Jazan University. The female nulliparous rats were kept separate (quarantined) from the male rats after birth. The polycarbonate (box) cage with fine rice husk was used as bedding to keep the animals. Before the experiment, the animals were kept for one week for acclimatization in the animal house of the College of Pharmacy in international standard laboratory environments, including a temperature of 25 ± 2 °C, a relative humidity of 45–55%, water ad libitum, and 12 h of light-dark cycles. These conditions were ideally maintained throughout the study regime.

Rats were given free access to ordinary pellet diets supplied by vendors and purchased from the local market. The experimental procedures complied with the ethical regulations of the Institutional Research Review and Ethical Committee (IRREC), Jazan University, Saudi Arabia (KSA), and studies were completed according to the standard guidelines of the Ethical Committee (Animal Ethical Approval Number: 723/216/1443). The food was withdrawn overnight (about 12 h) before the surgical procedure.

### 3.3. Experimental Design

The animals were randomly allocated (divided) into six groups, each with six rats. The first group used as a control was given saline (0.9 NaCl) orally for 15 days. The second group was given only CO (200 mg) orally for the same days as the control group, while the third group was treated once orally with APAP (2 g/kg b.w.). The fourth, fifth, and sixth groups were treated orally with varying (50, 100, and 200 mg/kg, respectively) doses of CO daily for 15 days and a single dose of APAP (2 g/kg) 3 days before completion. The selection of doses and dosing schemes were adapted according to our previous work [3,4,5]. On the 15th day, animals were anesthetized followed by sacrifice, and the uterus was quickly dissected out and kept at −80 °C for further investigation.

### 3.4. Uterus Tissue Sample Preparation

Uterus tissue homogenate (10% *w*/*v*) was prepared in child Tris-HCL (10 mM Tris-HCl pH 7.4) along with a cocktail of 10 μL/mL protease inhibitor. An Ultra Turrax T-25 (IKA, Staufen, Germany) homogenizer was used and set at a speed of 800× *g* for 5 min to homogenate. The homogenate was again centrifuged at 20,000× *g* for 20 min at 4 °C to separate the post-mitochondrial supernatant (PMS). Uterus tissue samples for the cytokine assay were prepared as described by the manufacturers (Abcam Kit, Cambridge, UK). 

### 3.5. Estimation of Lipid Peroxidation 

Lipid peroxidation was estimated with some modification by Hussain et al. [3]. The uterus tissue homogenate was used to estimate the content of lipid peroxidation in terms of MDA. For the MDA estimation assay, 250 µL of tissue homogenate was incubated at 37 °C and 0 °C. After 1 h, 0.5 mL of 10% trichloroacetic acid (TCA) and 0.5 mL of 0.67% thiobarbituric acid (TBA) were added to both sets of samples. The sample tube was centrifuged at 3000× *g* for 15 min, and the supernatant was positioned in a boiling water bath using a glass tube for 10 min. The absorbance was recorded at 535 nm after cooling. The LPO was estimated in nmoles of MDA/g tissue by a coefficient of 1.56 × 10^5^ M^−1^ cm^−1^.

### 3.6. Estimation of Reduced Glutathione (GSH) 

The GSH was estimated by Ashafaq et al. [5] using SSA (4%) in 1:1 with a sample (PMS). The PMS were precipitated with 0.1 mL of sulfosalicylic acid. The solution mixture was kept at 4 °C for 1 h and centrifuged at 3000 rpm for 15 min to separate the supernatant. The 3 mL assay sample includes 0.4 mL of DTNB, 2.2 mL of 0.1 M phosphate buffer (pH 7.4), and 0.4 mL of supernatant. The sample reading was taken instantly at 412 nm using a spectrophotometer (UV-1601, Shimadzu, Japan). A MEC of 13.6 × 10^3^ M^−1^cm^−1^ was used to determine the content of GSH as DTNB conjugate formed per mg protein.

### 3.7. Estimation of the Activity of the Superoxide Dismutase (SOD)

SOD activity was evaluated spectrophotometrically by observing the autooxidation of epinephrine as described earlier [27] at pH 10.4 for 3 min at 480 nm. The sample consists of 0.2 mL of uterine tissue PMS and 50 mM, pH 10.4, glycine buffer. The reaction was activated after adding epinephrine, and the activity of enzymes was calculated by using a molar extinction coefficient of 4.02 × 10^3^ M^−1^ cm^−1^. The optical density was taken at 480 nm. SOD is expressed as nmol of epinephrine protected from oxidation/min/mg protein.

### 3.8. Estimation of Catalase Activity 

Catalase activity was assessed according to Alshahrani et al. [5] with a 3 mL volume of the sample consisting of 1 mL (0.019 M) of H_2_O_2_, 50 μL (0.05 M) of PMS, and 1.95 mL of phosphate buffer. The reading of the assay sample was measured at 240 nm using the kinetic method. The activity of catalase is expressed as nmol H_2_O_2_ consumed/min/mg protein.

### 3.9. Assay of Glutathione Reductase (GR) and Glutathione Peroxidase (GPx) Activity 

GR activity was assayed by the method [28] as described by Ashfaq et al. (2017). The assay system consisted of 0.1 M PB pH 7.6, 0.1 mM NADPH, 0.5 mM EDTA, 1.0 mM GSSG, and 0.1 mL PMS in a total volume of 2.0 mL. The enzyme activity was quantitated at room temperature by measuring the disappearance of NADPH at 340 nm and was calculated as nmol NADPH oxidized/min/mg protein using a molar extinction coefficient of 6.22 × 10^3^ M^−1^ cm^−1^. GPx activity was measured according to the procedure in [28]. The reaction mixture consisted of 0.05 M PB pH 7.0, 1.0 mM EDTA, 1.0 mM sodium azide, 1.4 U of 0.1 mL GR, 1.0 mM glutathione, 0.2 mM NADPH, 0.25 mM hydrogen peroxide, and 0.1 mL of PMS in a final volume of 2.0 mL. The disappearance of NADPH at 340 nm was recorded at room temperature. The enzyme activity was calculated as nmol NADPH oxidized/min/mg protein using a molar extinction coefficient of 6.22 × 10^3^ M^−1^ cm^−1^.

### 3.10. Assay of Inflammatory Cytokine (IL-6 and IL-1 Beta) and Apoptosis Markers (Caspases 3 and 9)

Uterus tissue samples were homogenized in a buffer consisting of 20 mM Tris-HCl with pH 7.6, 100 mM KCl, 5 mM NaCl, 2 mM EDTA, 1 mM EGTA, 0.25 M sucrose, 2 mM DTT, and 2 mM PMSF with pH 7.2. Homogenates were centrifuged at 4 °C and 12,000 g for 15 min. Supernatants were removed and assayed in quadruplets to check the accuracy of results for the analysis of inflammatory cytokines (IL-1β and IL-6), and caspases 3 and 9 were estimated calorimetrically by using ELISA estimation kits (Abcam, UK). The procedures were followed as per the protocols provided by the manufacturer. A microplate reader (BioTek) was used to read the plate at 450 nm and 405 nm, respectively, according to the manufacturer’s guidelines (Abcam, UK). Tissue cytokine and caspase concentrations were expressed as picograms of antigen per mg of protein.

### 3.11. Western Blot Analysis of Caspases 3 and 9

Uterus tissue collected from all experimental groups was used for Western blot analysis. Nuclear extracts were prepared as described [26]. Samples containing equal amounts of protein were separated by 10–14% SDS–polyacrylamide gel electrophoresis. Proteins were transferred onto PVDF (polyvinylidene difluoride) membranes and incubated overnight at 4 °C with specific primary rabbit polyclonal antibodies against caspase 3 (dilution, 1:1000) and caspase 9 (dilution, 1:1000), followed by appropriate secondary antibodies conjugated to horseradish peroxidase. Protein bands were visualized using a Super Signal West Pico Kit (Thermo Fisher Scientific Pierce, IL, USA) according to the manufacturer’s instructions. All blots were stripped and re-incubated with a primary antibody specific to beta-actin (Sigma) as a loading control. The intensity of the bands was measured by densitometry and quantified using NIH-Image J software.

### 3.12. Uterus Histopathological Studies

Rats were transcardially perfused with physiological saline and then formalin. Uterus tissue was isolated from all group animals, cleaned using physiological saline (0.89%), and kept in a mixture of formaldehyde (40%), glacial acetic acid, and methanol (1:1:8, *v*/*v*) as a fixer for 48 h. Fixed tissue was embedded in beeswax paraffin, and tissue blocks were prepared. The 5 μm thick tissue sections of the uterus were collected from paraffin-embedded tissues. Further sections were processed through deparaffinization with the exchange of xylene, ethanol, and water. The deparaffinized sections were stained with hematoxylin and eosin, and sections were mounted to cover slip. Images were captured using a light microscope. The whole process is a slight modification of the protocol provided by Ashafaq et al. [28]. 

### 3.13. Estimation of Protein 

Protein was estimated by the well-known Lowry et al. [29] method using BSA as a standard.

### 3.14. Statistical Analysis

Results are expressed as mean ± SEM. The statistical analysis of the data was performed by applying the analysis of variance (ANOVA), followed by the Tukey–Kramer test. The *p* value of 0.05 was considered statistically significant. 

## 4. Discussion

In 2011, the USFDA mentioned that the dose of APAP must not exceed 4 g per day for adults, with the maximum being 325 mg in a single dose [30]. According to the Chinese Pharmacopoeia, the recommended dose of APAP should not be more than 2 g per single dose [31]. Recently, we have revealed the protective effect of CO against APAP toxicity in vivo using Wistar rats in different organs such as the liver, kidney, and brain, resulting in acute hepatotoxicity, nephrotoxicity, and neurotoxicity, to determine the protective effects of nutraceuticals [3,4,5]. Thus, this study was designed to investigate the possible protective effect of CO against APAP-induced uterine toxicity in Wistar rats.

Our current study showed consistency with the literature (14–16) and our previous studies (3–5), where APAP incites oxidative stress and defective antioxidative systems in several vital organs. In this study, there was an increase in the levels of LPO and reduced enzyme activity such as SOD, catalase, GR, GPx, and GSH in the APAP-treated group. This indicates that even a safe medication can be toxic when misused and suggests the necessity for an intervention to reverse such toxicity when it occurs. 

LPO is free radical-mediated cellular lipid damage whose intensity is increased during oxidative stress. Interestingly, CO significantly reduced the increased levels of LPO at the highest dose. This same dose had no effect on the non-APAP-treated group, indicating the specificity of CO against APAP-induced LPO expression. Moreover, there was a decrease in the enzymatic activity for the antioxidative enzymes such as SOD, catalase, GR, GPx, and the level of GSH in the APAP-treated group. However, animals co-treated with CO showed a marked regain of antioxidative enzyme activity and GSH contents in the APAP + CO group. The ability of CO to deplete free radicals and reduce oxidative stress was confirmed by the increase in the activities of antioxidative enzymes (SOD, catalase, GR, and GPx) and GSH, as well as the decrease in the level of LPO in the uterus tissue. This observation can be attributed to the strong antioxidative property of CO, as reported earlier in the liver [32], kidney [33], and brain [34]. Our results are consistent with those of other studies that made comparable observations on natural oils that had antioxidant properties [35,36]. 

APAP-induced uterus toxicity is correlated with inflammation because the activated metabolite of APAP (NAPQI) leads to the infiltration of inflammatory cells and overexpression of cytokines (such as IL-1β and IL-6), which culminate in inflammation and injury of the liver [37], kidney [38], and brain [39], released from 4 to 24 h after an overdose of APAP exposure [40]. In the present work, CO markedly slowed down the release of these cytokines caused by APAP overdose. This result represents a potential anti-inflammatory effect of CO against APAP-induced uterine toxicity. Therefore, the activity of CO is already reported by the liver [32], kidney [33], and brain [34]. Apoptosis in uterine cells is also due to APAP toxicity. Thus, suppressing apoptosis would reduce the development of toxicity. Many proteins involved in apoptosis, such as caspases 3 and 9, are upregulated in APAP-intoxicated cells. Thus, we examined the expression of caspases 3 and 9 in the uterine tissue. It was found that co-treatment with CO markedly ameliorated the expression of caspases 3 and 9 caused by APAP. These findings are supported by previous reports on the liver, kidney, and brain [32,33,34]. Furthermore, Western blot analysis also demonstrated similar results, confirming the protective role of CO in the prevention of APAP toxicity. 

Overdoses of APAP have been shown to induce apoptosis in several tissues and alter the deformity of the cells. Hence, we examined the possibility of APAP overdose-induced apoptosis in uterus cells in a histopathological experiment. As expected, APAP deformed the architecture of uterine tissue by degenerating epithelial cells and inducing apoptosis. This APAP toxicity was reversed dose-dependently by co-treatment with CO, where 100 mg only restored epithelial cells and 200 mg inhibited apoptosis. Thus, by suppressing apoptosis, the development of toxicity is reduced. 

Overall, CO’s protective effect against APAP-induced uterus toxicity is a promising strategy to reverse APAP-induced uterus toxicity and other organ toxicology, as reported earlier [3,4,5]. This work is consistent with our previous studies, in which we showed that APAP protects the liver, kidney, and brain from injury by generating free radicals and reducing antioxidant status [3,4,5]. Furthermore, we also showed that CO co-treatment protected the liver, kidney, and brain from injury by scavenging free radicals and improving antioxidative enzyme status. This could also explain the mechanism by which CO is mitigating APAP-induced uterine toxicity.

## 5. Conclusions

As a result of the current observation, it has been concluded that CO has antioxidative properties that contribute to its protective mechanism against APAP-induced uterus toxicity, which includes lowering LPO levels, increasing the activity of antioxidant enzymes, and regulating oxidative stress, the inflammatory response, and apoptosis.

## Figures and Tables

**Figure 1 plants-12-02290-f001:**
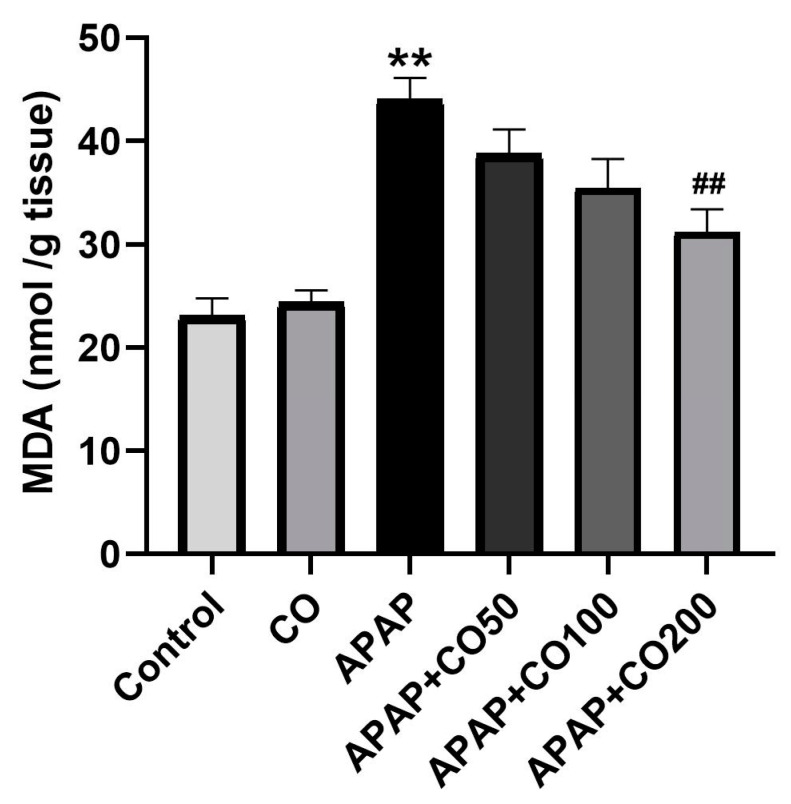
Effect of CO on uterine tissue levels of LPO in toxicity induced by APAP. Data presented as mean ± SEM (*n* = 6). ** *p* < 0.01 designates a significant difference between only APAP and the control group; ^##^ *p* < 0.01 shows a significant difference from the untreated APAP group (VEH).

**Figure 2 plants-12-02290-f002:**
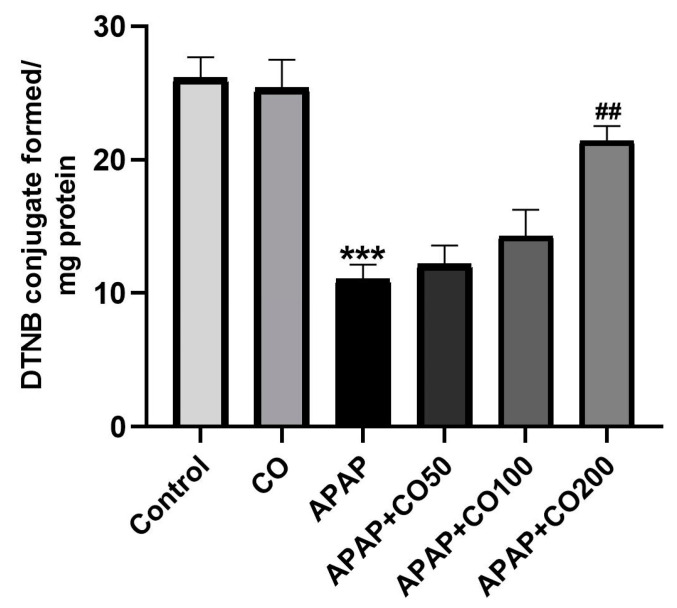
Effect of CO on uterine tissue levels of GSH in toxicity induced by APAP. Data presented as mean ± SEM (*n* = 6). *** *p* < 0.001 designates a significant difference between only APAP and the control group, and ^##^ *p* < 0.01 shows a significant difference from the untreated APAP group (VEH).

**Figure 3 plants-12-02290-f003:**
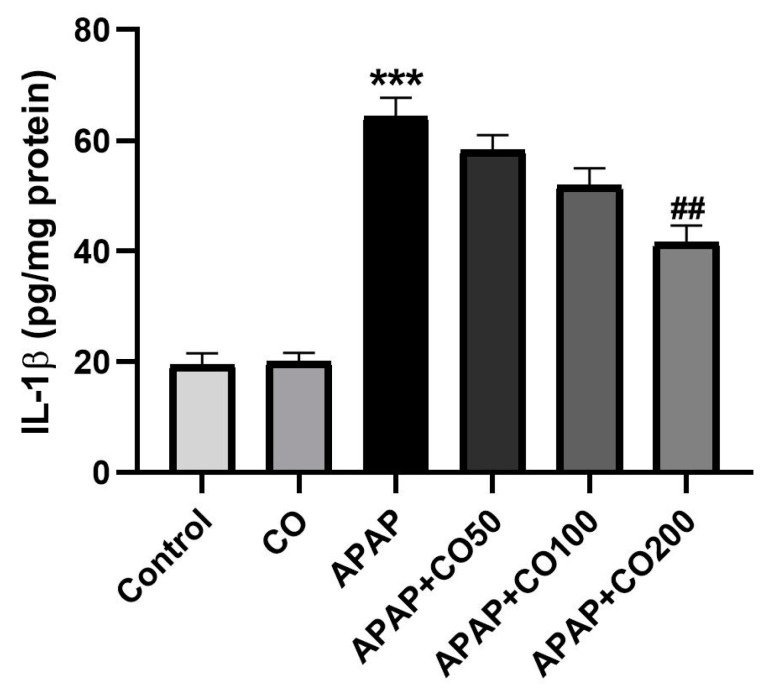
CO attenuates the APAP-induced level of IL-1β in the uterus of rats treated with APAP. ELISA results are presented as group mean ± SEM (*n* = 6). *** *p* < 0.001 compared to the control group; ^##^
*p* < 0.01 compared to the APAP group.

**Figure 4 plants-12-02290-f004:**
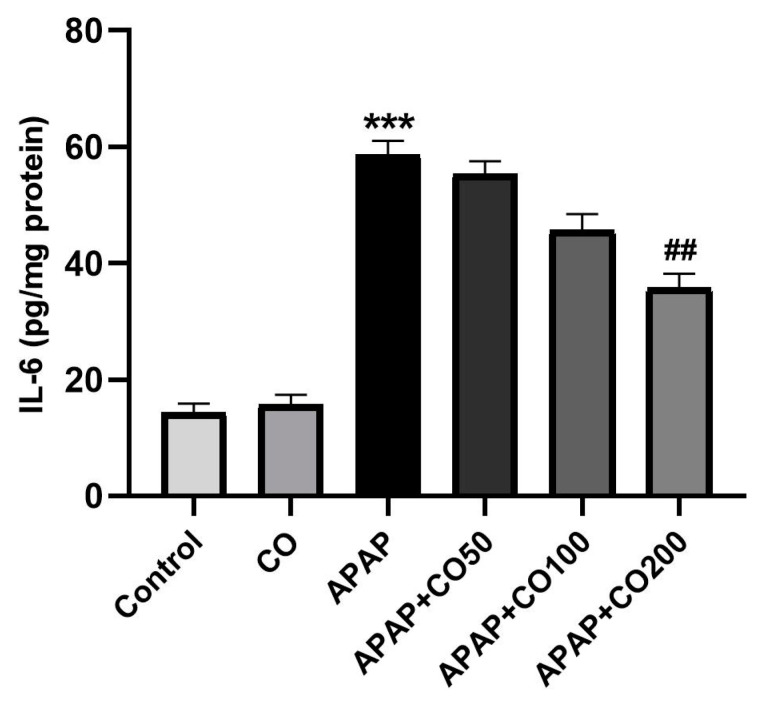
CO attenuates the level of IL-6 in the uterus of rats treated with APAP. The IL-6 ELISA assay is represented as the group mean ± SEM (*n* = 6). *** *p* < 0.001 compared to the control group; ^##^
*p* < 0.01 compared to the APAP group.

**Figure 5 plants-12-02290-f005:**
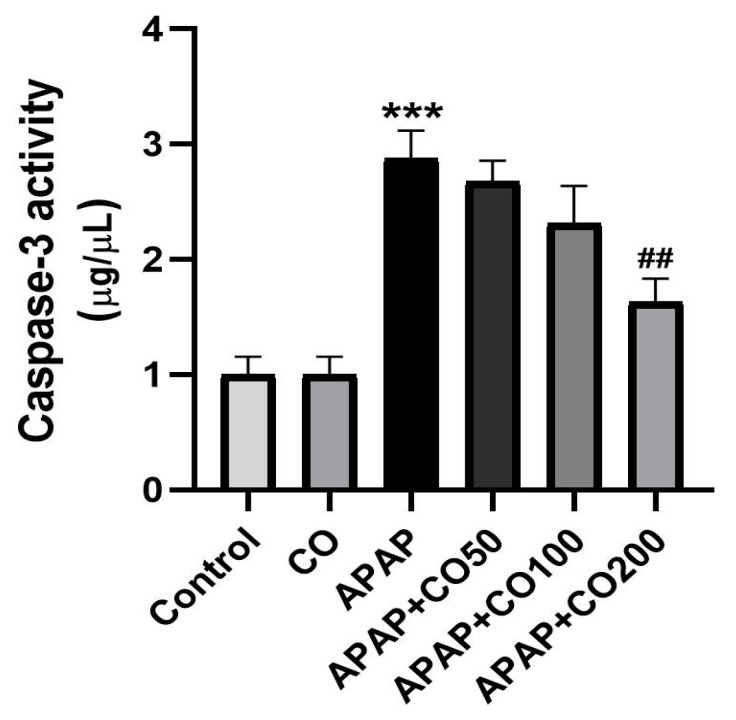
CO inhibits activation of caspase 3 in the uterine tissue of rats treated with APAP. Data are presented as mean ± SEM (*n* = 6). *** *p* < 0.001 designates a significant difference between APAP and the control group, and ^##^
*p* < 0.01 shows a significant difference from the APAP untreated group (VEH).

**Figure 6 plants-12-02290-f006:**
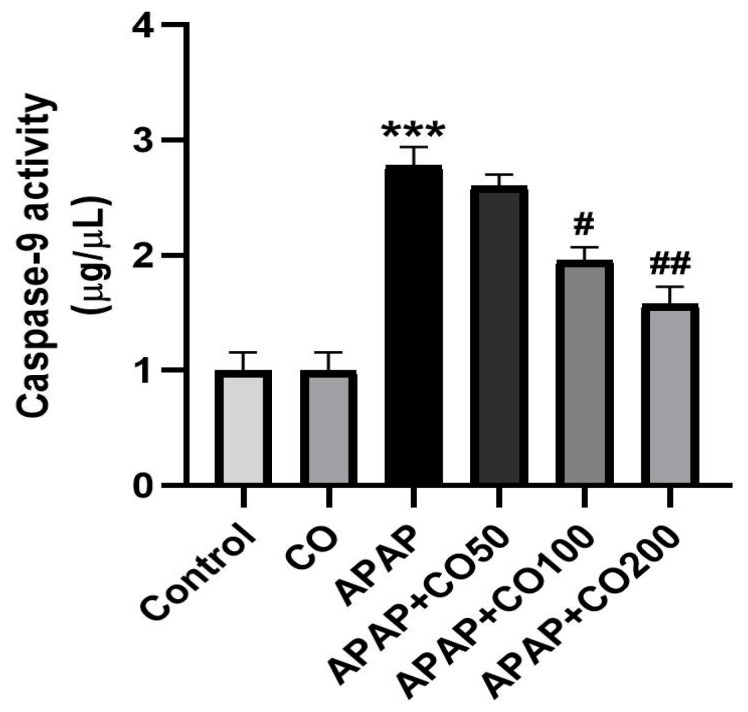
CO decreased the expression of caspase 9 in the uterine tissue of rats treated with APAP. Data presented as mean ± SEM (*n* = 6). *** *p* < 0.001 designates a significant difference between APAP and the control group, ^#^
*p* < 0.05, and ^##^
*p* < 0.01 show a significant difference from the APAP untreated group (VEH).

**Figure 7 plants-12-02290-f007:**
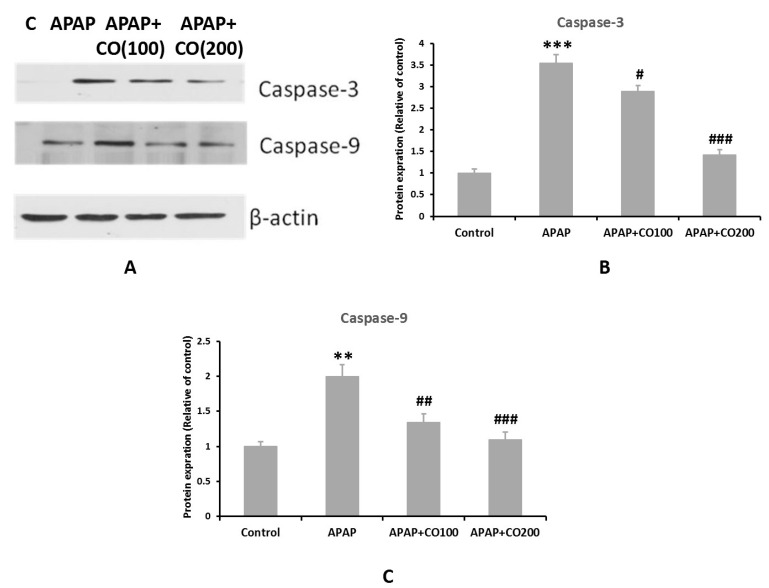
Western blot analysis of caspases 3 and 9 using SDS PAGE. ** *p* < 0.01, *** *p* < 0.001 designates a significant difference between APAP and the control group, ^#^ *p* < 0.05, ^##^ *p* < 0.01, and ^###^ *p* < 0.001 show a significant difference from the APAP Vs. APAP + CO 100 and APAP + CO 200.

**Figure 8 plants-12-02290-f008:**
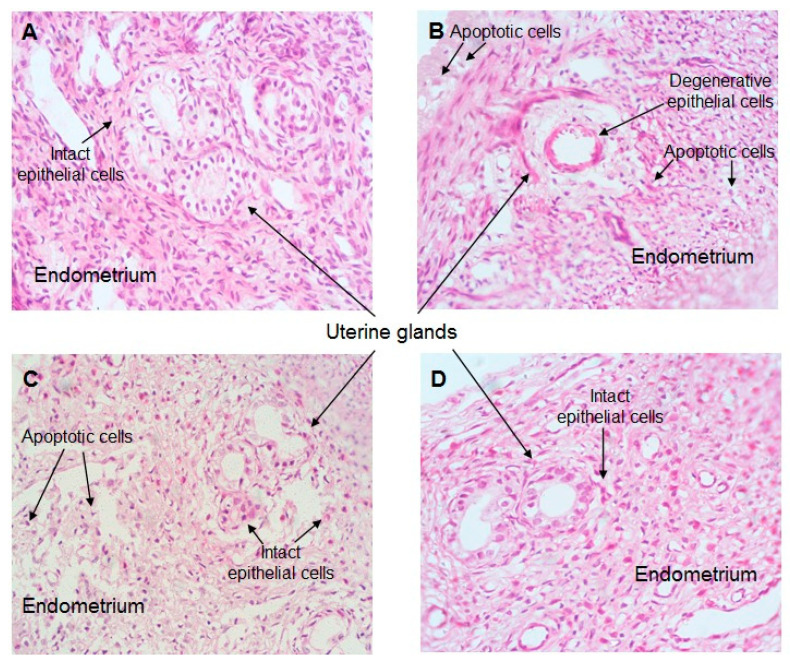
Histopathology of uterine tissue control (**A**), APAP only (**B**), APAP + 100 CO mg/dL (**C**), and APAP + 200 CO mg/dL (**D**).

**Table 1 plants-12-02290-t001:** Effect of CO on oxidative stress and antioxidant parameters (SOD, CAT, GPx, and GR) in uterine tissue.

Parameters	Control	CO Only	APAP Only	APAP + CO 50	APAP + CO 100	APAP + CO 200
SOD (nmol of epinephrine protected from oxidation/min/mg protein)	250.27 ± 24	249.93 ± 19 (−0.13%)	108.62 ± 16 *** (−56.59%) ^a^	115.33 ± 17 (6.17%) ^b^	156.57 ± 19 (44.14%) ^b^	200.66 ± 22 ^##^ (84.73%) ^b^
CAT (nmol H_2_O_2_ consumed/min/mg protein)	14.26 ± 1.23	15.02 ± 1.06 (5.32%)	8.62 ± 1.06 ** (−39.55%) ^a^	10.37 ± 0.91 (25.54%) ^b^	12.85 ± 1.72 ^#^ (55.13%) ^b^	13.02 ± 1.87 ^##^ (57.62%) ^b^
GPx (nmol of NADPH oxidized/min/mg protein)	159.03 ± 17	157.87 ± 19 (−0.72%)	67.79 ± 15 *** (−57.37%) ^a^	92.73 ± 11 (36.79%) ^b^	98.66 ± 13 ^##^ (45.53%) ^b^	122.41 ± 18 ^###^ (80.57%) ^b^
GR (nmol of NADPH oxidized/min/mg protein)	200.82 ± 24	203.11 ± 17 (1.14%)	109.39 ± 16 *** (−45.52%) ^a^	118.09 ± 19 (7.95%) ^b^	125.28 ± 15 (14.65%) ^b^	171.79 ± 21 ^###^ (57.04%) ^b^

APAP treatment significantly decreased antioxidant enzymes in the APAP group as compared to the control. However, CO treatment significantly attenuated the activity of antioxidant enzymes in the APAP + CO group as compared to APAP. Values are expressed as the mean ± SEM of *n* = 6 animals. ^a^ Values in parentheses indicate the percentage change vs. control. ^b^ Values in parentheses indicate the percentage change vs. APAP. ** *p* < 0.01, *** *p* < 0.001 APAP vs. control. ^#^
*p* < 0.05, ^##^
*p* < 0.01, ^###^
*p* < 0.001 APAP + CO vs. APAP.

## Data Availability

The data available within the articles.

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
