# Peer review of "Cinnamon Oil Alleviates Acetaminophen-Induced Uterine Toxicity in Rats by Abrogation of Oxidative Stress, Apoptosis, and Inflammation"

_plants, 2023, doi:10.3390/plants12122290_

Round 1

Reviewer 1 Report

plants-2259087

Cinnamon Oil Alleviates Acetaminophen Induced Uterus Toxicity in Wistar Rats by Abrogation of Oxidative Stress, Apoptosis and Inflammation

By Sohail Hussain, Saeed Alshahrani , Rahimullah Siddiqui , Andleeb Khan , Manal Mohamed Elhassan Taha , Rayan A. Ahmed , Abdulmajeed M. Jali , Marwa Qadri , Mohammad Ashafaq.

In this study Hussain et al evaluated the protective effects of different doses of cinnamon oil against toxicity induced by paracetamol on uterus tissue of female rats. They observed that only the highest administered dosage of cinnamon oil (200mg/kg) was effective against toxicity caused by oxidative stress and reduced inflammation and apoptosis. This is an interesting work that explores new effects of cinnamon oil. However there are some flaws that must be solved.

The graphical abstract does not clearly depict the effects of cinnamon oil in the different processes.

The introduction in too long and should be summarized.

In the materials and methods section, in GSH estimation please specify what kind of sample was used.

In uterus histopathological studies, what does “perfused uterus tissue” mean? The methods should be better described.

Figures 5 and 6, the activity of caspases should be expressed as xmol/unit of time/ ug protein.

The title of 3.6 section must be rephrased. The optical density of bands of caspases must be quantified and normalized against beta-actin bands, to assess and state that there were significant differences in protein expression. In figure 7 the MW should be added. Original images of whole blots should be displayed.

Minor comments

Please, standardize numbers of references in the main text, some of them are in round brackets other in square brackets.

Author Response

Dear Reviewer 

We greatly appreciate you taking the time to examine our paper; it will help us produce better scientific research.

Please find the response to the comment attached here.

Thanks 

Reviewer 2 Report

The Manuscript "Cinnamon Oil Alleviates Acetaminophen-Induced Uterus Toxicity in Rats by Abrogation of Oxidative Stress, Apoptosis and Inflammation" explored the Acetaminophen-induced inflammation and oxidative stress, which augment cell death markers in rat uterus and the recuperative effect of Cinnamon Oil on these stresses. Unfortunately, the assays performed appear to be a loose collection of the recuperative impact of CO on rat uterus without validation experiments confirming potential mechanisms. It is entirely unclear how specific those changes are as the author showed similar effects on rats' liver, kidney, and brain as it might lead to many of the same changes no matter which tissue is used. Overall, this is a weak study that needs a lot of new experiments (mechanistic studies) to prove the point. Lack the mechanism of action and reporting just restoration of the broad phenomenons in the rat uterus tissue.

Minor comments

1.     At the beginning of the discussion section, the author mentioned the US-FDA suggested dose of APAP must not exceed 4g per day for adults, and many studies conducted on rats used 100-300mg/kg body weight. How did the author determine the dose of APAP 2g/kg bw on a single day? What is the rationale behind using APAP multiple fold high of an optimal dose? Isn't such a high dose cause a general cytotoxic effect on vital organs?

2.     The author mentioned in each figure legend, "Data presented as Mean ± SEM (n=6)" If all the animals in a group (6) were used for Uterus tissue sample preparation for biochemical assays, how did the author perform histopathological assay?

3.     The Discussion lacks coherence in many places and merely reports the finding of this study. It required significant rewriting and discussion with the author's conclusions and interpretation concerning previously published studies.

4.     Properly label Figure 7

5.     Check for grammatical and spelling mistakes (Line 173, 259)

Author Response

(The authors gave the same response as above.)

Reviewer 3 Report

The manuscript “Cinnamon Oil Alleviates Acetaminophen -Induced Uterus Toxicity in Rats by Abrogation of Oxidative Stress, Apoptosis and Inflammation” have reported the significant effects of Cinnamon Oil against Acetaminophen induced Uterus toxicity. The authors have used several techniques to prove their hypothesis. After thoroughly reviewing I feel the manuscript NO needs to be revise prior to acceptance.

Comment

1.        Abbreviations should be defined at first mention and used consistently thereafter. Give the full form for all the abbreviations during their first appearance.

2.        Materials are reported in a very succinct manner. Revised the oxidative stress markers assay protocol in method section.

3.         Histopathology method should be explained in detail.

4.        The conclusion should be revised.

5.        English language of the manuscript should be improved and the minor errors should be corrected.

6.        Please write the catalogue no of kits used in the manuscript.

7.        Make the changes in the language of Abstract.

8.        Include some recent references for the beneficial effect of cinnamon in uterus, if available.

Author Response

(The authors gave the same response as above.)

Round 2

Reviewer 1 Report

Authors addressed all the reviewer’s comments and improved the manuscript as suggested. There is no further comment.

Reviewer 2 Report

None